# Reliability of surface electromyographic (sEMG) measures of equine axial and appendicular muscles during overground trot

L. St. George[1]*, T. J. P. Spoormakers[2], S. H. Roy[3], S. J. Hobbs[1], H. M. Clayton[4], J. Richards[5], F. M. Serra Bragança[2]

1 Centre for Applied Sport and Exercise Sciences, University of Central Lancashire, Preston, Lancashire, United Kingdom, 2 Department of Clinical Sciences, Faculty of Veterinary Medicine, Utrecht University, Utrecht, The Netherlands, 3 Delsys/Altec Inc., Natick, Massachusetts, United States of America, 4 Sport Horse Science, Mason, Michigan, United States of America, 5 Allied Health Research Unit, University of Central Lancashire, Preston, Lancashire, United Kingdom

* lbstgeorge@uclan.ac.uk

**Data Availability Statement:** All relevant data are within the paper and its Supporting Information files.

## Abstract

The reliability of surface electromyography (sEMG) has not been adequately demonstrated in the equine literature and is an essential consideration as a methodology for application in clinical gait analysis. This observational study investigated within-session, intra-subject (stride-to-stride) and inter-subject reliability, and between-session reliability of normalised sEMG activity profiles, from triceps brachii (triceps), latissimus dorsi (latissimus), longissimus dorsi (longissimus), biceps femoris (biceps), superficial gluteal (gluteal) and semitendinosus muscles in n = 8 clinically non-lame horses during in-hand trot. sEMG sensors were bilaterally located on muscles to collect data during two test sessions (session 1 and 2) with a minimum 24-hour interval. Raw sEMG signals from ten trot strides per horse and session were DC-offset removed, high-pass filtered (40 Hz), full-wave rectified, and low-pass filtered (25 Hz). Signals were normalised to peak amplitude and percent stride before calculating intra- and inter-subject ensemble average sEMG profiles across strides for each muscle and session. sEMG profiles were assessed using waveform similarity statistics: the coefficient of variation (CV) to assess intra- and inter-subject reliability and the adjusted coefficient of multiple correlation (CMC) to evaluate between-session reliability. Across muscles, CV data revealed that intra-horse sEMG profiles within- and between-sessions were comparatively more reliable than inter-horse profiles. Bilateral gluteal, semitendinosus, triceps and longissimus (at T14 and L1) and right biceps showed excellent between-session reliability with group-averaged CMCs > 0.90 (range 0.90–0.97). Bilateral latissimus and left biceps showed good between-session reliability with group-averaged CMCs > 0.75 (range 0.78–0.88). sEMG profiles can reliably describe fundamental muscle activity patterns for selected equine muscles within a test session for individual horses (intra-subject). However, these profiles are more variable across horses (inter-subject) and between sessions (between-session reliability), suggesting that it is reasonable to use sEMG to objectively monitor the intra-individual activity of these muscles across multiple gait evaluation sessions at in-hand trot.

**Funding:** This study was made possible with support from Morris Animal Foundation (https://www.morrisanimalfoundation.org/) (Grant ID: D21EQ-406) and the British Society of Animal Science (https://bsas.org.uk/) 2018 Steve Bishop Early Career Award, both awarded to LSG. The funders had no role in study design, data collection and analysis, decision to publish, or preparation of the manuscript.

**Competing interests:** I have read the journal's policy and the authors of this manuscript have the following competing interests: author SR is employed by Delsys Inc., the manufacturers of the sEMG sensors used in this study. The remaining authors declare that no competing interests exist. This does not alter our adherence to PLOS ONE policies on sharing data and materials.

## Introduction

Surface electromyography (sEMG) is increasingly used to study coordination among superficial musculature during "normal" or "non-lame" equine locomotion [1–11]. sEMG activation patterns of various appendicular and axial muscles have been described for horses during non-lame walk, trot, and canter [2–11], based on several temporal and amplitude-based sEMG variables. More recently, sEMG has been used to study lameness in horses [12–14], based on adaptations in muscle activity. However, researchers have not yet conducted a study to determine whether so-called "normal" profiles of sEMG activity can be reliably identified during equine gait, taking into consideration the variability observed within- or between test sessions. sEMG measures must be reliable if they are to eventually be applied in practice to aid clinical diagnostics and/or monitoring of treatment techniques for horses. Further, understanding the between-session reliability of equine sEMG profiles is essential for evaluating treatment or intervention effects when using repeated measures [15].

In human studies, the evaluation of pathological gait is often based on a comparison between experimental (pathological) and control (non-pathological) groups, where normative sEMG activation profiles across control subjects are implicitly assumed and provide a template for comparative analyses of effect size [16]. In horses, only one comparative gait study of sEMG data between groups of non-lame and lame horses is known [12]. This study made the following implicit assumption described by Arsenault et al. [16]: during non-lame walking and trotting, horses exhibit repeatable and indistinguishable differences in the activation pattern of superficial muscles, which can therefore be used as a reference for describing adaptations during lameness. However, this assumption of so-called "normal" muscle activation patterns has not been verified for horses using empirical evidence and in large enough cohorts. Indeed, biological variation in the axial [17–19] and appendicular [20, 21] movement of non-lame horses has been reported during walk and trot, where intra-horse (stride-to-stride) variability is generally reported as being comparatively lower than inter-horse variability. Thus, it is necessary to determine whether there is also a normative range of variability for the muscle activation patterns of clinically non-lame horses.

During human locomotion, low levels of intra-subject variability, alongside comparatively greater levels of inter-subject variability have generally been reported for sEMG profiles, suggesting that individual differences occur within muscle recruitment profiles, which vary from subtle to extreme, depending on the muscle studied [16, 22–25]. These individual manifestations are generally lost when sEMG data are pooled across subjects [16]. Although it is generally accepted that sEMG is stable within a single data collection session (intra-subject reliability), the re-application of electrodes causes some degree of variation within sEMG signals between test sessions [26–29]. Still, the reliability of sEMG profiles from the human literature are generally in accordance with acceptable standards for gait analysis within clinical and research settings [28]. As such, the American Academy of Neurology recommends sEMG as a clinical tool for the kinesiologic analysis of movement disorders [30]. We suggest that sEMG holds similar promise for use in clinical equine gait analysis [13, 14].

To our knowledge, only a few equine gait studies have described the reliability or variation observed for sEMG data, collected during treadmill locomotion [5–7, 31]. As such, the reliability of sEMG profiles from a healthy population of horses remains largely unknown during over-ground trot. This represents a gap in knowledge, particularly given the known biomechanical differences between treadmill and over-ground trot in horses [32], with the latter generally employed for clinical gait analysis. The current study aims to determine whether a typical profile of sEMG activity can be reliably described within- and between horses for selected appendicular and axial superficial muscles during in-hand trot. A secondary aim is to

determine the between-session reliability of these profiles. Based on previous sEMG literature from human and equine subjects, it was hypothesised that reliable activity profiles can describe fundamental muscle activity patterns for selected equine muscles within a test session for individual horses (intra-subject reliability), but that these profiles will be more variable across horses (inter-subject reliability) and even more variable between test sessions (between-session reliability).

## Materials and methods

Ethical approval for this study was obtained from Utrecht University (CCD: AVD108002015307) and the University of Central Lancashire (Reference number: RE/17/08a_b). The study was conducted at the Equine Department of Clinical Sciences of Utrecht University, where sEMG (2000 Hz) and three-dimensional (3D) motion capture data (200 Hz) were synchronously collected from horses during in-hand trot trials, conducted on a straight, hard-surfaced indoor runway. Each horse underwent two separate data collection sessions (session 1 and session 2), which were conducted on separate days, with a minimum period of 24 hours and a maximum period of 48 hours between sessions.

### Horses

Eight (n = 8) horses (sex: 7 mares, 1 stallion, age: 9.2 ± 3.9 years, height: 161.3 ± 3.4 cm, body mass: 582.1 ± 39.4 kg, breed: 7 Dutch Warmblood, 1 Friesian) from Utrecht University's herd were used. Horses were in regular use for low-level dressage and pleasure riding. No lameness was observed during visual examination by two experienced equine practitioners (T.S., F.S.B.) prior to data collection at walk and trot on a straight line. Horses were housed at the Equine Department of Clinical Sciences of Utrecht University in standard large box stalls during the experimental period (4 days) and received daily group turn out in an outdoor paddock.

### Instrumentation and equipment set up

To collect sEMG and three-dimensional (3D) motion capture data, horses were instrumented with sEMG sensors and retro-reflective markers in accordance with the methods described by St. George et al. [14] and Spoormakers et al. [13].

### Surface electromyography (sEMG)

Wireless sEMG sensors (Delsys Trigno, Delsys Inc., USA) were positioned to record bilaterally from the following superficial muscles: longissimus dorsi (longissimus), long head of triceps brachii (triceps), latissimus dorsi (latissimus), superficial gluteal (gluteal), vertebral head of biceps femoris (biceps) and semitendinosus. The reader is referred to St. George et al. [14] and Spoormakers et al. [13] for detailed descriptions of sensor site locations for the appendicular and axial muscles, respectively. Once sensor locations were determined, hair was removed from each site using clippers (No. 40 clipper blade) and then the skin was thoroughly cleaned using isopropyl alcohol. A small amount of saline solution was applied to each electrode bar to act as an electrolytic solution [33, 34] and sensors were then attached over the middle of the muscle belly, with the electrodes oriented perpendicular to the underlying muscle fibre direction [35, 36], as determined using ultrasonography. Sensors were attached to the skin using Delsys Adhesive Surface Interface strips (Delsys Inc., USA), combined with double-sided tape, attached to the top and bottom of the sensor, above each electrode pair. A drop of cyanoacrylate glue was also placed on top of the double-sided tape, above each electrode pair. To minimize sensor re-application errors during test session 2, sensor locations were marked on the

horse's skin using permanent marker during test session 1 and a single, experienced researcher performed all sensor applications (L.S.G).

## Kinematics

An optical motion capture (OMC) system with eighteen high-speed, infrared cameras (Oqus 700+, Qualisys AB, Sweden) was used to collect 3D kinematic data. Cameras were secured to the walls of a large indoor hall, where veterinary lameness examinations are conducted. The system was calibrated for each data collection session and produced an extended calibration volume approximately 56 m long and 10 m wide. The OMC system was hardware synchronised to the sEMG system to record both time series in one file for further processing. To collect 3D kinematic data, retro-reflective markers (19 mm diameter super-spherical markers, Qualisys AB, Sweden) were attached over anatomical landmarks on the forelimbs, hindlimbs and back, as described in [13, 14]. Hair was clipped from each location to ensure optimal adhesion and consistent placement across data collection sessions.

## Data acquisition protocol

sEMG (2000 Hz) and 3D kinematic (200 Hz) data were synchronously collected using Qualisys Track Manager (Qualisys AB, Sweden) software, as the horse trotted over the runway four times, twice in each direction. One handler led all of the horses throughout the data collection sessions and permitted them to trot at their preferred speed. Baseline data were initially collected during each test session, then mild, reversible lameness was induced for the purposes of another study [13, 14], where details of the lameness induction protocol can be found. Only baseline data from each test session were employed for this study and horses were deemed as clinically non-lame (< 1/5 AAEP Lameness Scale) during session 1 and session 2 through visual, clinical assessments by two qualified veterinarians (T.S., F.S.B). Objective lameness assessment was also undertaken using motion asymmetry parameters, calculated using kinematic data [13, 14].

## Data processing

Kinematic data were tracked in Qualisys Track Manager and imported into Visual3D (Version 2021.06.2, c-Motion Inc., USA) and Matlab (Version 2020b, TheMathWorks Inc., USA) software for further analysis. The detection of hindlimb impact events, used for stride segmentation, were conducted in Matlab, in accordance with the method described by Roepstorff et al. [37], using the maximal vertical displacement of the marker placed between the tubera sacrale. These events were manually imported into Visual3D for stride segmentation of sEMG data. As left and right-side muscles were analysed separately, contralateral hindlimb impact events were employed for stride segmentation of sEMG signals. To evaluate the phasic activity of sEMG profiles in relation to motion profiles during trot, sagittal plane joint angles that each of the studied appendicular muscles work on (shoulder, elbow, hip, stifle and tarsal joints) as well as overall fore- and hindlimb pro-retraction angles, were calculated in Visual3D, in accordance with the methods described by St. George et al. [14]. Thoracolumbar flexion/extension and lateral bending angles were also calculated in Visual3D using cranial and caudal segments, defined using markers located on the T6 and T13 vertebrae, and on the T13 vertebra and the tuber sacrale, respectively, as described by Spoormakers et al. [13].

Raw sEMG signals were differentially amplified by a factor gain of 909, a common-mode rejection ratio (CMRR) of >80 dB and an internal Butterworth high-pass (20 ± 5 Hz cut-off, >40 dB/dec) and low-pass filter (450 ± 50 Hz cut-off, >80 dB/dec). Post-processing and analysis of sEMG signals was conducted in Visual3D and included DC-offset removal, followed by

the application of a high-pass filter (Butterworth 4th order, 40 Hz cut-off) to attenuate low-frequency noise contamination [17], and then full-wave rectification. For each horse, high-pass filtered, and full-wave rectified sEMG signals were enveloped using a Butterworth 4th order, low-pass filter (25 Hz cut-off) and were normalised to a reference voluntary contraction (RVC): the peak amplitude value of enveloped signals observed for each muscle location across all included trot strides from the corresponding test session [4]. To ensure that signals were normalised to a peak value that accurately reflects muscular effort, outliers in peak amplitude data were detected prior to the normalisation of continuous signals [14, 38].

In accordance with previous studies on the reliability of sEMG profiles [16, 39], data were reduced so that 10 strides from each horse and test session (session 1 and 2) were selected for further analysis. Data were first reduced by removing outlier strides that were detected using sEMG data from each muscle, based on the method described by St. George et al. [14, 38], and excluded from further analysis. Stride velocity was employed for further data reduction, given the known impact of gait speed on sEMG variability [8, 40]. Stride velocity was calculated in Matlab using the smoothed differentiation of the horizontal coordinates (x, y) of the reflective marker between the tubera sacrale. For each horse, mean stride velocity was calculated across strides within each test session, and the strides with the greatest deviation from the mean value were excluded to produce ten strides for further analysis. The selected strides were time normalised to 0–100% of the stride cycle (101 data points per stride) in Visual3D. Then, intra- and inter-subject ensemble sEMG profiles were calculated for each muscle in Visual3D, by averaging time and amplitude-normalised, and linear enveloped sEMG signals across the selected strides from each horse, muscle, and test session. Ensemble, between-session sEMG profiles were also calculated for each horse and muscle in Visual3D by averaging sEMG signals from the included strides across test sessions (session 1 and 2).

## Statistical analysis

Waveform similarity statistics were employed to evaluate the reliability of sEMG ensemble profiles from each muscle. The coefficient of variation (CV) and the coefficient of multiple correlation (CMC) were calculated given their frequent use in studies of human sEMG reliability [15, 23, 26, 39, 41–44] and in some studies of equine axial and appendicular kinematics [18, 20, 45], enabling comparison of results with other studies, and because reporting multiple reliability statistics is recommended [15, 46]. The CV was calculated in accordance with Winter [47, 48], to quantify the intra- (stride-to-stride) and inter-subject reliability of sEMG profiles within each test session [22, 23, 42, 44]. For continuous sEMG profiles, the CV was calculated as the root mean square of the standard deviation (RMSD) of the stride period, divided by the mean ensemble average (intra- or inter-subject average) over the stride [47, 48]. When calculated this way, the CV can be considered a measure of the variability-to-signal ratio for sEMG data from each muscle [22] and tends toward zero when waveforms are similar. Intra-subject CV values from each horse, muscle, and test session were averaged to create summary statistics (mean ± standard deviation) across the sample.

The CMC was calculated in accordance with Kadaba et al. [26], to quantify the between-session reliability of intra-subject sEMG ensemble profiles from each muscle using Matlab. CMCs from each horse and muscle were averaged to create summary between-session statistics (mean ± standard deviation) across the sample. Interpretation of CMC values was based on following the convention that *moderate reliability* is between 0.50 and 0.75, *good reliability* is between 0.75 and 0.90, and *excellent reliability* >0.90 [39]. To enable direct comparisons of within- and between-session reliability, we calculated CVs for each muscle using intra- and inter-subject average sEMG profiles, calculated using strides from both test sessions [15, 26].

## Results

A total of 152 and 147 trot strides were included in the analysis of left (session 1: 78, session 2: 74) and right (session 1: 80, session 2: 67) muscles, respectively. For the included trot strides, mean ± standard deviation (SD) stride velocity was 3.09 ± 0.29 m/s (session 1: 3.11 ± 0.23 m/s, session 2: 3.07 ± 0.35 m/s) and stride duration was 0.75 ± 0.03 s (session 1: 0.74 ± 0.03 s, session 2: 0.75 ± 0.03 s).

### Within-session reliability of intra- and inter-subject sEMG profiles

Across test sessions and muscles, intra-subject CVs ranged between 0.41 for left longissimus (L1 location) and 0.83 for right biceps, with inter-subject CVs ranging from 0.59 for right longissimus (L1 location) to 1.08 for right biceps (Table 1). Reliability within a test session was better for intra-subject sEMG profiles, compared to inter-subject sEMG profiles and this was true across all muscles and test sessions (Table 1). Longissimus (at T14 and L1) displayed the lowest intra-subject (CV range: 0.41–0.54) and inter-subject (CV range: 0.59–0.83) variability, with the biceps and semitendinosus displaying the highest variability (intra-subject CV range: 0.62–0.83, inter-subject CV range: 0.84–1.08) within test sessions (Table 1). Intra-subject sEMG profiles from a representative subject (horse 4) and inter-subject sEMG profiles from session 1 are shown in Figs 1 and 2, respectively, along with the corresponding CVs depicting within-session reliability. For comparative purposes, intra-subject sEMG profiles from the same representative subject, as well as inter-subject profiles from session 2 are presented in S1 and S2 Figs, respectively. Intra- and inter-subject sEMG profiles and CV data from each horse, muscle and test session are presented in S1 Table and S3–S18 Figs.

### Between-session reliability of intra- and inter-subject sEMG profiles

Bilateral gluteal, semitendinosus, triceps and longissimus (at T14 and L1) and right biceps showed excellent between-session reliability with group-averaged CMCs > 0.90 (range 0.90–0.97) (Table 2). Bilateral latissimus and left biceps showed good reliability with group-averaged

**Table 1. Inter-subject coefficients of variation (CV) and mean and standard deviation (SD) intra-subject CVs, calculated across (n = 8) horses, for within-session (session 1 and session 2) sEMG profiles from selected superficial muscles.**

| Muscle | Side | Session 1 | | | Session 2 | | |
|---|---|---|---|---|---|---|---|
| | | Intra-Subject CV | | Inter-Subject CV | Intra-Subject CV | | Inter-Subject CV |
| | | Mean | SD | | Mean | SD | |
| Triceps brachii | Left | 0.70 | 0.09 | 0.94 | 0.62 | 0.16 | 0.83 |
| | Right | 0.65 | 0.13 | 0.93 | 0.73 | 0.23 | 0.90 |
| Latissimus dorsi | Left | 0.58 | 0.10 | 0.81 | 0.55 | 0.11 | 0.80 |
| | Right | 0.51 | 0.10 | 0.79 | 0.55 | 0.08 | 0.72 |
| Longissimus T14 | Left | 0.42 | 0.09 | 0.67 | 0.54 | 0.21 | 0.83 |
| | Right | 0.52 | 0.15 | 0.73 | 0.45 | 0.06 | 0.69 |
| Longissimus L1 | Left | 0.41 | 0.06 | 0.60 | 0.41 | 0.08 | 0.64 |
| | Right | 0.42 | 0.10 | 0.59 | 0.48 | 0.18 | 0.69 |
| Superficial gluteal | Left | 0.53 | 0.04 | 0.77 | 0.54 | 0.09 | 0.84 |
| | Right | 0.56 | 0.05 | 0.85 | 0.56 | 0.11 | 0.86 |
| Biceps femoris | Left | 0.69 | 0.10 | 0.90 | 0.67 | 0.12 | 1.00 |
| | Right | 0.67 | 0.10 | 0.84 | 0.83 | 0.22 | 1.08 |
| Semitendinosus | Left | 0.67 | 0.12 | 0.97 | 0.62 | 0.17 | 0.89 |
| | Right | 0.72 | 0.16 | 0.87 | 0.66 | 0.12 | 0.93 |

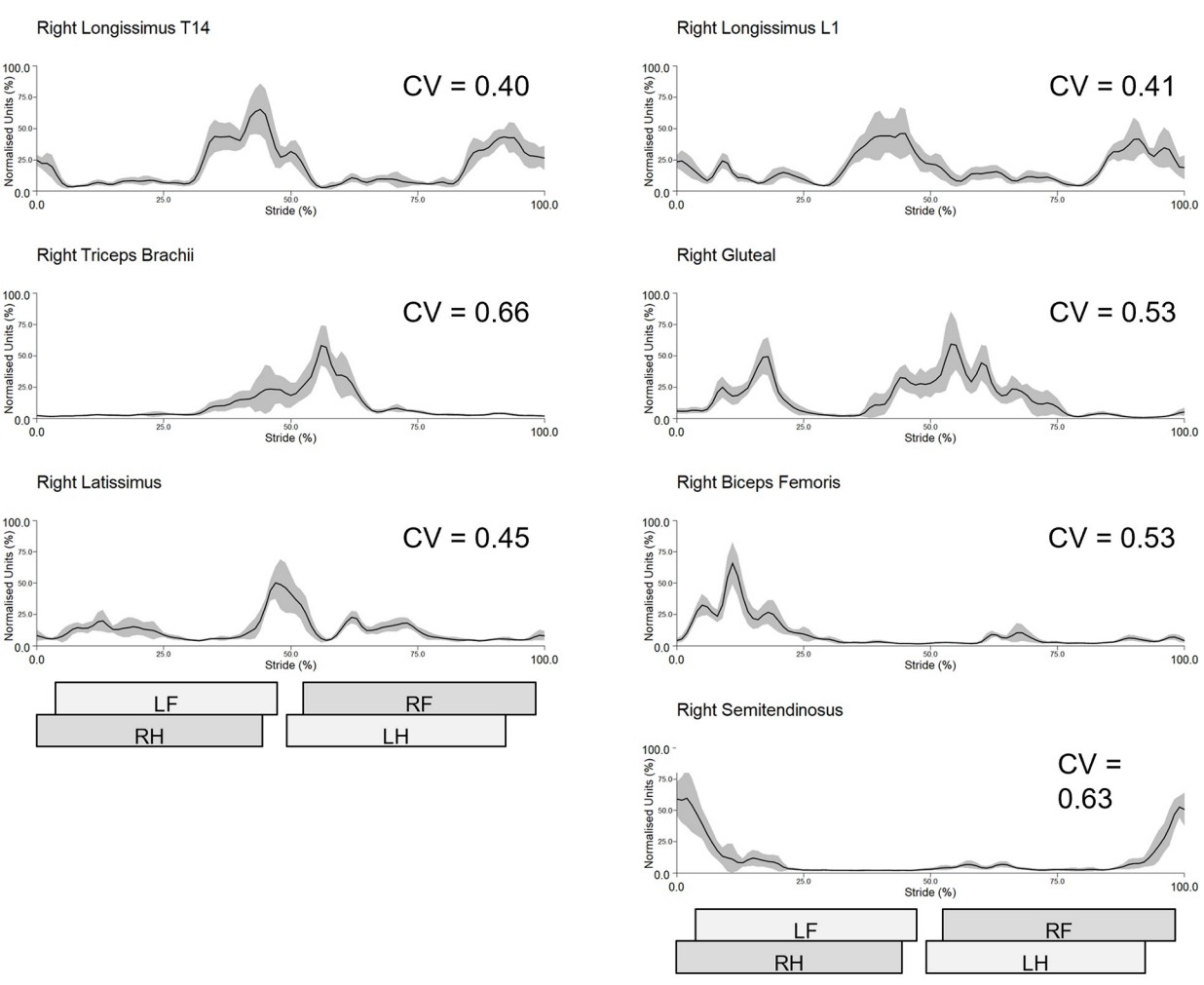

**Fig 1. Within-session (session 1) intra-subject sEMG profiles from the right-side muscles of a representative subject (horse 4).** Mean (solid line) and standard deviation (grey shaded area) time and amplitude-normalised sEMG data from 10 trot strides are presented. Coefficient of variation (CV) is indicated for each muscle. Stance durations for left and right forelimb (LF, RF) and hindlimb (LH, RH) are presented as horizontal bars, giving a temporal reference to the trot stride cycle.

CMCs > 0.75 (range 0.78–0.88) (Table 2). Between-session intra-subject sEMG profiles from a representative subject (horse 4) and inter-subject sEMG profiles are shown in Figs 3 and 4, respectively, along with the corresponding CVs and CMCs for each muscle. To enable interpretation of muscle activity profiles in the context of motion profiles during trot, Fig 4 also includes continuous angle-time curves from the joints/segments that each of the studied muscles work on. Between-session intra- and inter-subject sEMG profiles, CV, and CMC data from each horse and muscle are presented as Supporting Information (S1 and S2 Tables, S19–S27 Figs).

Mean and SD intra-subject CV, and inter-subject CV, depicting reliability between test sessions are presented in Table 3. Intra-subject between-session CVs ranged between 0.48 for left longissimus (L1 location) and 0.86 for right biceps, with inter-subject CVs ranging from 0.62 to 0.98 for the same muscles. The reliability between test sessions was better for intra-individual sEMG profiles, compared to inter-individual sEMG profiles across all muscles and test sessions. Longissimus (at T14 and L1) displayed the lowest between-session intra-subject (CV

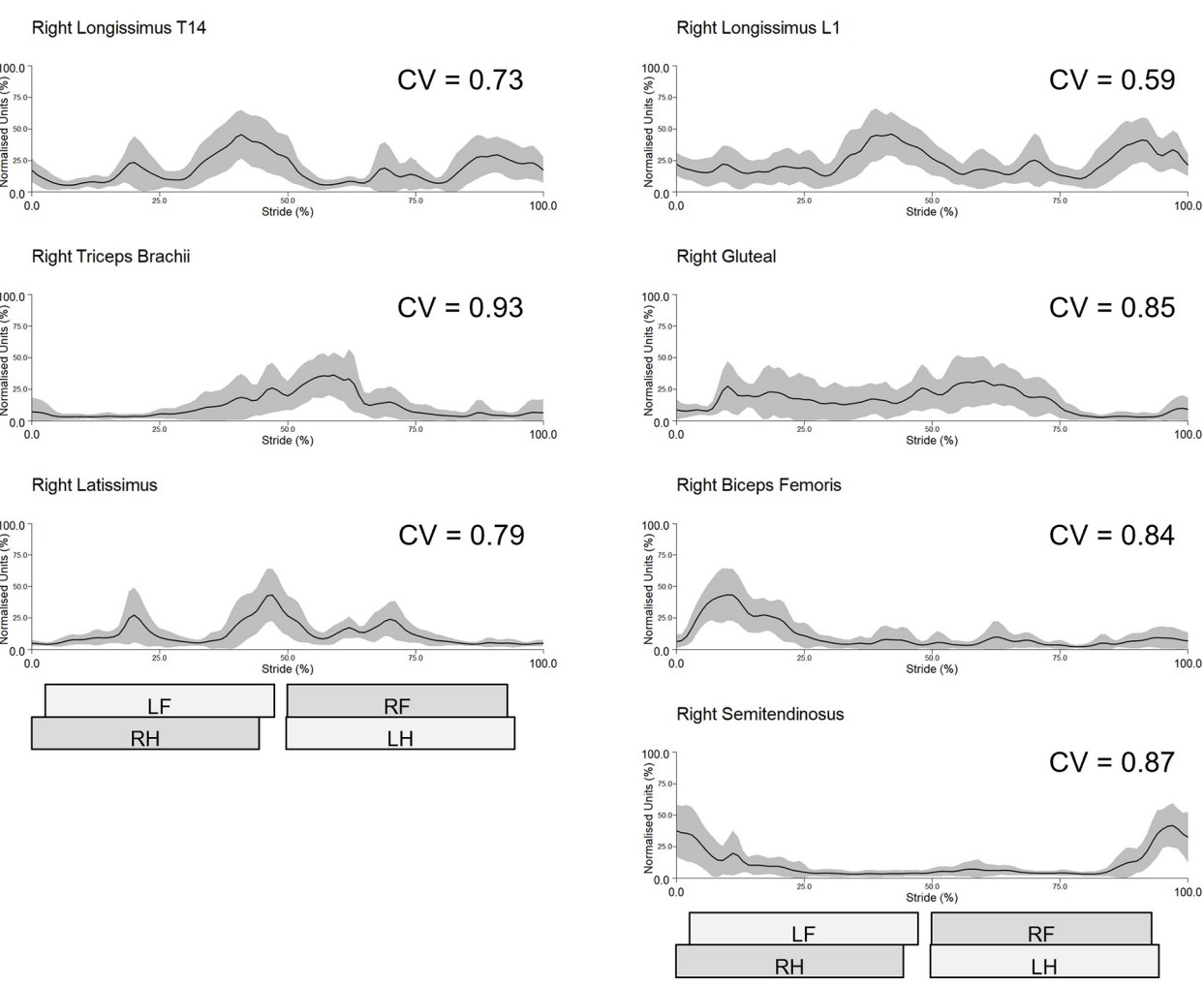

**Fig 2. Within-session (session 1) inter-subject sEMG profiles from the right-side muscles across the group of horses (n = 8).** Mean (solid line) and standard deviation (grey shaded area) time and amplitude-normalised sEMG data from 80 trot strides are presented. Coefficient of variation (CV) is indicated for each muscle. Stance durations for left and right forelimb (LF, RF) and hindlimb (LH, RH) are presented as horizontal bars, giving a temporal reference to the trot stride cycle.

range: 0.48–0.59) and inter-subject (CV range: 0.64–0.76) variability, with the biceps and semitendinosus displaying the highest variability (intra-subject CV range: 0.74–0.86, intra-subject CV range: 0.91–0.98), which agrees with the within-session findings. Across muscles, mean intra-subject CVs show that between-session sEMG profiles were more variable (Table 3), compared to their corresponding within-session profiles (Table 1), except for the right gluteal, which had a between-session intra-subject CV of 0.54 and within-session CV of 0.56 for sessions 1 and 2.

## Discussion

In this study, we provide the first comprehensive evaluation of intra- and inter-horse reliability, both within- and between test sessions, of sEMG profiles obtained from a range of appendicular and axial superficial muscles during in-hand, straight line trot. To fulfil the study aim, we employed waveform similarity statistics, specifically CV and CMC, that have been employed in similar studies evaluating the reliability of sEMG and kinematic profiles during

**Table 2. Mean and standard deviation (SD) of the intra-subject coefficient of multiple correlation (CMC), calculated across horses (n = 8) and test sessions (session 1 and session 2) for selected superficial muscles.**

| Muscle | Side | Intra-Subject CMC | |
|---|---|---|---|
| | | Mean | SD |
| Triceps brachii | Left | 0.95 | 0.04 |
| | Right | 0.97 | 0.04 |
| Latissimus dorsi | Left | 0.88 | 0.13 |
| | Right | 0.87 | 0.21 |
| Longissimus T14 | Left | 0.97 | 0.05 |
| | Right | 0.97 | 0.02 |
| Longissimus L1 | Left | 0.91 | 0.08 |
| | Right | 0.93 | 0.10 |
| Superficial gluteal | Left | 0.96 | 0.03 |
| | Right | 0.90 | 0.05 |
| Biceps femoris | Left | 0.78 | 0.24 |
| | Right | 0.97 | 0.03 |
| Semitendinosus | Left | 0.96 | 0.07 |
| | Right | 0.96 | 0.04 |

equine and human gait [15, 18, 20, 22, 23, 26, 39, 42, 44, 45]. For within-session reliability, our CV results showed that intra-subject sEMG profiles were less variable than inter-subject profiles. We measured between-session reliability using CMC and found that 11 of 14 studied muscles showed excellent between-session reliability (CMC > 0.90), with the remaining three muscles showing good reliability (CMC > 0.75) across the group of horses. As a direct comparison, we also measured between-session reliability using CV and found that intra-subject profiles were more variable between testsessions than between strides (within-session), and this was true for 13 of the 14 muscles studied. Thus, our results support our hypothesis that reliable sEMG activity profiles can describe fundamental muscle activity patterns for selected equine muscles within a test session for individual horses (intra-subject) and that these profiles will be more variable between horses (inter-subject) and test sessions (between-session reliability).

## Within-session reliability of intra- and inter-subject sEMG profiles during trot

The coefficient of variation was employed to measure intra- and inter-subject reliability of sEMG profiles within each test session, as it has been validated in the human and equine literature, as an effective means of analysing the waveform similarity of sEMG, kinematic, and kinetic data over the gait cycle [20, 26, 45, 49]. For studies employing CV to measure the reliability of discrete sEMG variables, thresholds of < 0.12 or < 0.15 have been described as respectively indicating "acceptable" or "good" reliability [41, 50]. However, to our knowledge, no such thresholds have been described for the evaluation of continuous waveforms using CV in either human or equine literature. Instead, researchers have relied on comparisons with other studies to interpret the reliability of sEMG or kinematic profiles during gait [22, 23, 26, 43, 44]. In keeping with this convention, the CV values that were observed here for intra-subject sEMG profiles across all studied muscles, generally fell within the range of values reported in the literature for sEMG profiles during non-pathological human gait [22, 23, 26, 42–44]. In these studies, it was concluded that intra-individual sEMG profiles were "highly repeatable" and "extremely stable" [16, 23, 26], based on the measured CVs. As this is the first known

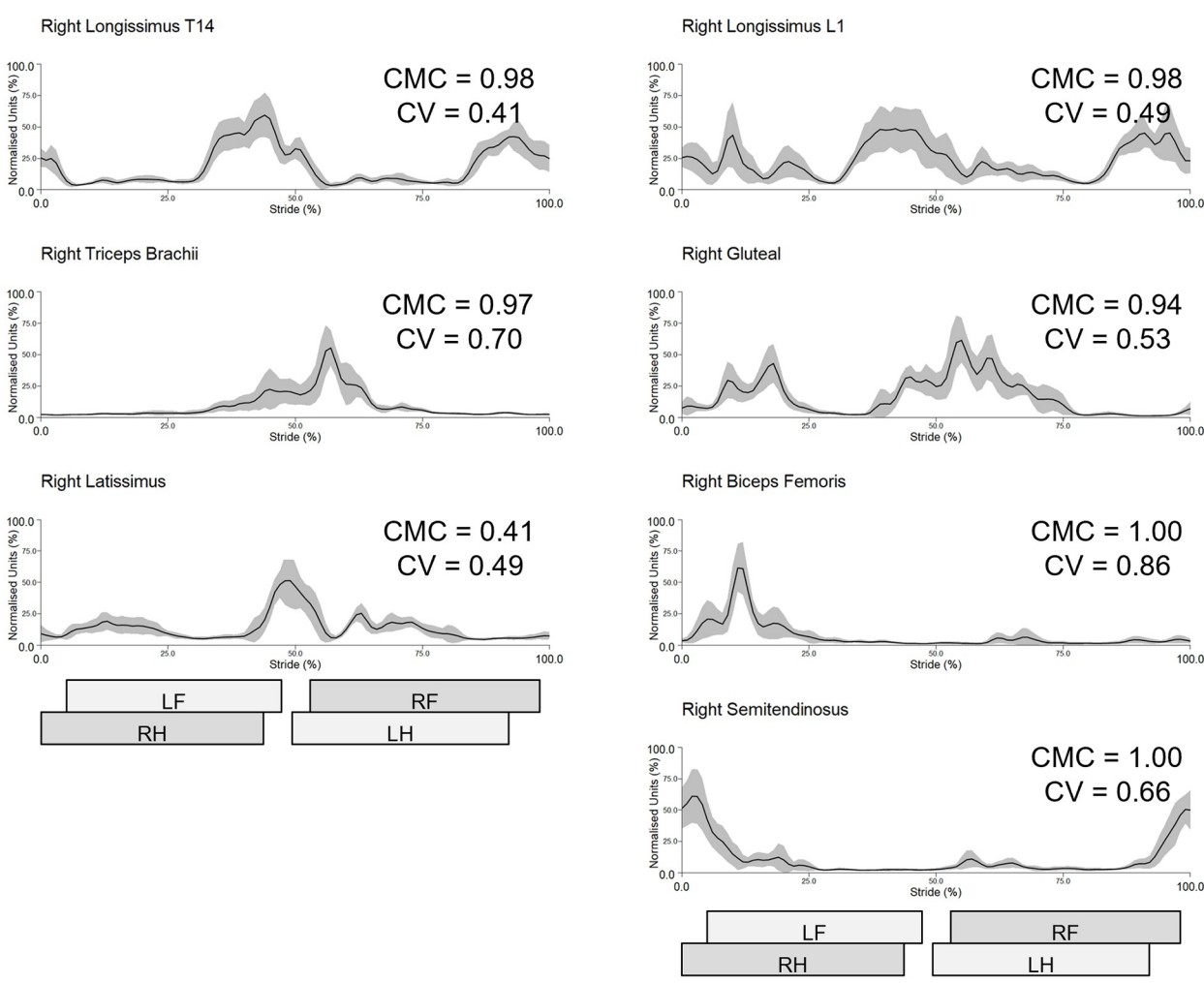

**Fig 3. Between-session, intra-subject sEMG profiles from the right-side muscles of a representative subject (horse 4).** Mean (solid line) and standard deviation (grey shaded area) time and amplitude-normalised sEMG data from 18 trot strides are presented. Coefficient of variation (CV) and coefficient of multiple correlation (CMC) are indicated for each muscle. Stance durations for left and right forelimb (LF, RF) and hindlimb (LH, RH) are presented as horizontal bars, giving a temporal reference to the trot stride cycle.

study to report CV values for equine sEMG profiles, it seems reasonable to adopt the same interpretation of CV results from the human literature within the context of sEMG reliability, particularly given the unique challenges that come with reducing sources of variability to acquire high-fidelity sEMG signals from horses [1, 51].

The lowest intra- and inter-subject CVs were observed for longissimus, at both T14 and L1 locations, suggesting that this muscle displays the least variability in comparison to the other appendicular muscles studied here. After longissimus, the lowest variability in sEMG intra-subject profiles was observed for latissimus and gluteal, with CVs ranging from 0.51–0.58. In contrast, the bi-articular semitendinosus, biceps and triceps muscles exhibited the highest CVs across muscles and sessions. Interestingly, this finding agrees with previous studies that have observed greater variability for the phasic activity profiles of bi-articular proximal limb muscles during human walking and running, compared to mono-articular limb muscles [16, 22, 25, 26, 39, 44]. This is possibly because EMG data from bi-articular muscles are affected by the degree and velocity of movement across two joints [25], or because these proximal muscles

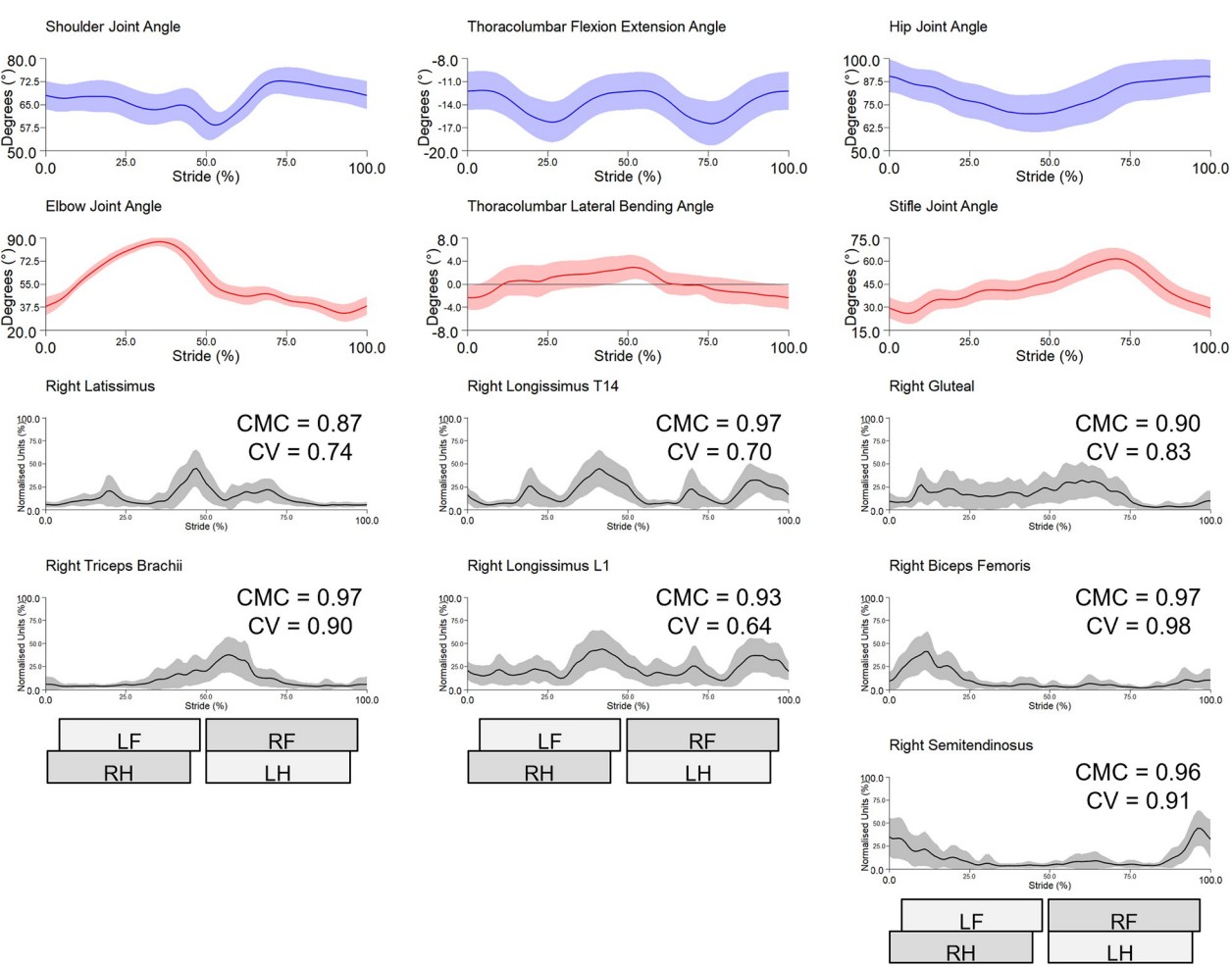

**Fig 4. Between-session, inter-subject sEMG profiles from right-side muscles across the group of horses (n = 8) and test sessions (session 1 and 2).**
Mean (solid black line) and standard deviation (SD) (grey shaded area) time and amplitude-normalised sEMG data from 147 trot strides are presented. Coefficient of variation (CV) and coefficient of multiple correlation (CMC) are indicated for each muscle. Within each panel, mean (blue and red solid lines) and SD (blue and red shaded areas) sagittal plane joint/segment angles from corresponding trot strides are presented alongside the muscles that work on them. Stance durations for left and right forelimb (LF, RF) and hindlimb (LH, RH) are presented as horizontal bars, giving a temporal reference to the trot stride cycle.

exhibit greater adaptability for producing gait [22]. Inter-muscle variability may also be related to lower force impulses, as a result of body movement, that are experienced at the longissimus sensor sites compared to the appendicular muscle sites. These differences may result in comparatively lower levels of mechanical perturbation to the electrode-skin interface and subsequent variability withins sEMG signals from the longissimus locations [52]. Although differences in muscle function and sensor location may offer one explanation for differences in inter-muscle variability, one must also consider the nature of the CV calculation when interpreting these differences. sEMG profiles with lower mean activity levels or increased areas of quiescent baseline activity may be prone to higher CVs [22, 43, 49, 53]. Indeed, we observed that the muscles with the highest intra- and inter-individual CVs (triceps, biceps, semitendinosus) generally exhibited one main burst of activity, from late swing to mid-stance phase, with quiescent baseline activity for the remainder of the stride cycle. However, we did observe individual manifestations for these muscles, with some intra-subject profiles displaying an inconsistent burst of activity during late swing phase, which may account for some of the measured

**Table 3. Inter-subject coefficients of variation (CV) and mean and standard deviation (SD) intra-subject CVs calculated across (n = 8) horses and test sessions (session 1 and session 2) for selected superficial muscles.**

| Muscle | Side | Intra-Subject CV | | Inter-Subject CV |
|---|---|---|---|---|
| | | Mean | SD | |
| Triceps brachii | Left | 0.74 | 0.13 | 0.90 |
| | Right | 0.73 | 0.10 | 0.90 |
| Latissimus dorsi | Left | 0.61 | 0.11 | 0.81 |
| | Right | 0.58 | 0.11 | 0.74 |
| Longissimus T14 | Left | 0.59 | 0.17 | 0.76 |
| | Right | 0.55 | 0.12 | 0.70 |
| Longissimus L1 | Left | 0.48 | 0.08 | 0.62 |
| | Right | 0.52 | 0.08 | 0.64 |
| Superficial gluteal | Left | 0.60 | 0.09 | 0.81 |
| | Right | 0.54 | 0.09 | 0.83 |
| Biceps femoris | Left | 0.78 | 0.09 | 0.95 |
| | Right | 0.86 | 0.17 | 0.98 |
| Semitendinosus | Left | 0.78 | 0.15 | 0.94 |
| | Right | 0.74 | 0.11 | 0.91 |

variability. In comparison, the muscles with the lowest CVs (longissimus, latissimus, gluteal) exhibited comparatively higher baseline activity, alongside longer and/or multiple activation bursts throughout the stride cycle. Despite its sensitivity to the mean value of the sEMG signal, the measured CVs for the studied muscles fell within acceptable limits described in the literature [16, 23, 26], suggesting that individual horses display a consistent phasic activation pattern within a test session, and that these patterns are the least variable for the longissimus and the most variable for the studied bi-articular fore- and hindlimb muscles.

As hypothesised, inter-horse sEMG profiles displayed higher variability, as quantified by higher CV values, when compared to intra-horse sEMG profiles across muscles. Kinematic studies have observed comparatively lower biological variability within horses, than between horses for axial and appendicular movement [17–21] and our findings suggest that the same is true for the underlying muscle activation that facilitates these movements. Faber et al. [45] observed that CVs were 2–3 times higher for inter-horse than intra-horse axial movement patterns during walk. Interestingly, we did not observe the same disparity between inter- and intra-horse CV values for sEMG profiles, but our inter-subject CVs ranged from 0.59 to 1.08, which are generally higher, but fall within the range of CV values reported by Faber et al. [45] for axial movement profiles. This may suggest that sEMG profiles generally exhibit greater variability than movement profiles, but that there is comparatively less variability between inter- and intra-horse sEMG profiles. Further studies are required to confirm this. In contrast to intra-subject sEMG profiles, variability in inter-subject profiles may be influenced by internal and external factors that are beyond the control of the experimenter. These factors include, but are not limited to between-subject differences in subcutaneous fat, motor unit recruitment patterns, skin temperature, skin impedance, electrode positioning, and subject cadence [22, 42]. Our findings agree with previous studies of human locomotion that have reported higher CVs for inter-subject sEMG profiles when compared to individual sEMG profiles [16, 22–25]. As such, it has been suggested that group-averaged sEMG profiles provide an average functional range of activation for that muscle during gait, which should be interpreted as the thresholds for between-subject differences rather than a true pattern of activation [16]. Our findings imply that the same is true for equine sEMG profiles and further work is required to establish

thresholds for acceptable levels of inter-individual variation within the wider equine population. It is from here that normative ranges of EMG activity profiles can be established for specific muscles during equine gait.

## Between-session reliability of intra- and inter-subject sEMG profiles during trot

Between-session reliability of intra-horse sEMG profiles was quantified using the CMC measure and the mean CMC values that were observed here were similar to those reported in the literature for human sEMG profiles during walking and running gait, [26, 39] and for axial movement profiles during equine walk and trot on a treadmill [18]. The lowest between-session reliability was observed for the left biceps muscle. In contrast, the right biceps displayed the highest mean CMC, alongside longissimus (at T14) and right triceps. Although bilateral sEMG data were acquired from the studied muscles, it was not our intention to evaluate bilateral symmetry or reliability of sEMG activity profiles. However, it is interesting to note that, apart from biceps, the mean difference between left and right CVs and CMCs within each muscle was ≤ 0.14, which indicates comparable levels of bilateral reliability within the studied muscles during trot. However, further research is required to confirm this. Bilateral differences observed for biceps CMC may be related to differences in sensor placement, laterality, movement asymmetry or a combination of these factors and this should also be explored in future research.

Unlike within-session measures of reliability, between-session reliability is susceptible to sensor re-application errors [26, 39], where even minor placement differences have been shown to significantly affect phasic activity patterns of sEMG signals [54]. We attempted to minimise re-application errors by following standardized protocols of marking sensor locations on the skin with permanent marker and by having a single, experienced researcher perform all sensor applications. Despite our efforts, it is not possible to ensure that the same volume of muscle was measured during different test sessions [26] and this will inevitably account for some of the variability in sEMG signals between sessions. Still, sensor re-application is inherent for monitoring patient progress using clinical gait analysis [55], and so our results exhibit external validity and suggest that sEMG profiles from selected appendicular and axial muscles exhibit high levels of between-session reliability for in-hand trot, when measured using CMC.

The CMC has been used as a measure of within-day reliability for human sEMG profiles, but this calculation is reliant on sEMG data from separate, stand-alone gait trials within a given day [26, 49]. In this study, sEMG and kinematic data were continuously collected within one session, where horses traversed the runway four consecutive times during in-hand trot with the same handler. Thus, sEMG data from stand-alone gait trials were not collected, so we did not deem it appropriate to calculate within-session CMC, as the lack of variation between each runway crossing would likely produce misleadingly low levels of variation. As such we used CV as an additional measure of between-session reliability, which enabled direct comparisons of within- and between-session reliability. Between-session CVs ranged between 0.48–0.86 within horses and 0.62–0.98 between horses. According to the CV values observed here, intra-subject reliability within a test session was consistently better than between test sessions across the studied muscles, which agrees with human and equine biomechanics studies that have made the same comparison [26, 29, 55]. The order of reliability was the same within- and between testsessions, with longissimus profiles displaying the highest reliability, followed by latissimus and gluteal, and with the bi-articular muscles displaying the lowest reliability, as measured using both CV and CMC. Kadaba et al. [26] reported greater reliability for sEMG

profiles during human walking when measured using CMC, compared to CV [26], which was also observed here. This can be explained by the fact that these two measures do not measure reliability in the same way. The CV is a measure of absolute reliability where variance between strides is normalised to the mean, and its magnitude is therefore unbounded [27, 41]. Whereas the CMC calculation normalises variation between strides to the total variance, and the ratio is scaled from 0–1 making it an attractive measure of waveform similarity [27]. As there doesn't appear to be a consensus on what CV values constitute acceptable reliability for continuous data, we defer to the CMC values in our conclusion that sEMG profiles are highly repeatable between test sessions. However, as between-session CVs fell within the same range as those reported in the human literature, we consider this to be an indication of acceptable between-session reliability.

## Study limitations

A relatively small sample of eight horses was employed for this study, which can be considered a limitation for the extrapolation of findings to the wider equine population. The horses were deemed as being clinically non-lame, based on subjective and objective veterinary assessments. However, it is important to note that the sEMG and kinematic data employed in this study formed part of a larger, novel dataset where an acute lameness induction model was employed to study adaptations in muscle activity and movement [13, 14]. Thus, it is possible that the lameness induced on session 1 could be a potential source of variability between sessions. However, studies on the general/ridden horse population have shown that substantial movement asymmetries/lameness is present in up to 75% of horses in regular work [56, 57], so it may not be possible to fully eradicate this source of variability from future studies.

Another controllable factor is gait speed, which we did not standardise. This could be considered a limitation, as gait speed effects the amplitude and phasic activation patterns of sEMG signals [40] and can be standardised in horses using high-speed treadmills. However, studies on the reliability of sEMG signals have reported that, within a human subject, the preferred walking speed produces a more reproduceable sEMG signal, than when speed is controlled, which requires conscious effort [27, 41, 44, 50, 54]. Future studies are required to confirm whether this is the case for horses, but we felt it was important for this preliminary investigation to emulate real-world clinical gait analysis conditions for horses, where overground trot is employed. It is important to note that we attempted to reduce between-subject and session variability in gait speed by employing the same handler for all in-hand trot trials and observed only a small difference of 0.04 m/s in mean trot stride velocity between test sessions.

The choice of signal processing methods, smoothing and normalisation, are a source of variability that is within the control of the experimenter [39, 42–44, 58]. It has been shown that the intra-individual variability of sEMG waveforms is reduced by applying a greater degree of smoothing during post-processing, [26, 27, 39, 58, 59]. For example, Kleissen et al. [58] observed intra-subject CVs of 0.70 vs. 0.31 when a low-pass filter of 25 Hz and 3 Hz cut-off were respectively applied to sEMG profiles from the gluteus medius of human subjects during self-selected walking speed. Signals were smoothed here using a low-pass filter with a 25 Hz cut-off frequency, which has been employed in previous equine and human research as this retains individual fluctuations in amplitude and phasic activation patterns within EMG signals [13, 14, 38, 58]. A comparison of different signal filtering methods was beyond the scope of this study and should therefore be evaluated in future studies.

The choice of normalisation method has been widely reported to influence the intra- and inter-subject variability of sEMG profiles [39, 42–44]. Normalisation to the within-subject peak amplitude, as employed here, or mean amplitude of the dynamic sEMG signal, has been

shown to reduce variability during human walking and running, particularly when compared to un-normalised signals or signals normalised to a maximum voluntary contraction (MVC) [39, 42–44]. Normalisation to an MVC is recommended where the goal is to evaluate the degree of muscle activity that is required to undertake dynamic tasks [43]. It is not possible to obtain an MVC from equine subjects and it was not our goal to evaluate the level of muscle activation, but rather to ascertain whether templates for EMG activity during trot can be reliably measured. Thus, we are confident in our decision to normalise using peak amplitude but suggest that future studies investigate the effect of other normalisation methods, namely the mean dynamic method, as this may offer guidance on reducing variability in sEMG profiles for equine gait analysis.

Finally, a lack of comparative studies on the reliability of equine sEMG profiles meant that we often drew from the human literature when interpreting our findings. We endeavoured to limit these comparisons to the general interpretation of CV and CMC values related to the reliability of sEMG waveforms during gait. However, this can be considered a limiting factor of our study, particularly as there are several sources of variability that may confound inter-species comparisons of sEMG data, for example: differences in subcutaneous fat and skin impedance, as well as differences in the body weight, speed and gait characteristics of horses, which result in greater impact forces and resultant perturbations to the electrode-skin interface [51, 60]. Thus, further studies are required to enable direct comparisons of findings and to determine the normative range of variability, as measured using CV and CMC, for the muscle activation patterns of clinically non-lame horses.

## Conclusions

Across all studied muscles, intra-subject sEMG profiles showed the highest reliability, particularly within a test session, suggesting that individual horses exhibit a stable pattern of muscle function during in-hand trot on a straight line. sEMG profiles were found to be more variable between horses (inter-subject reliability) and test sessions (between-session reliability), both of which may be partially explained by biological variation between horses, with the latter being uniquely and inherently affected by sensor re-application errors. Measures of between-session reliability, using CMC, showed excellent or good reliability across all studied muscles, suggesting that it is reasonable to use sEMG to objectively monitor the activity of these muscles across multiple gait evaluation sessions at trot. This study offers a first step in determining whether there is a normative range of variability for the muscle activation patterns of clinically non-lame horses. Thus, future research should build on this preliminary study to explore, and eventually validate, normative sEMG profiles for equine gait, so that they may be used as a reference to aid clinical decision-making for detecting and monitoring gait abnormalities in horses.

## Supporting information

**S1 Table. Within- and between-session intra-subject coefficients of variation (CV) from each of the studied horses (n = 8), muscles and test sessions (session 1 and session 2).** (PDF)

**S2 Table. Between-session, intra-subject coefficient of multiple correlation (CMC) from each of the studied horses (n = 8) and muscles, calculated across test sessions (session 1 and session 2).** (PDF)

**S1 Fig. Within-session (session 2) intra-subject sEMG profiles from the left and right-side muscles of horse 4.** Mean (solid line) and standard deviation (grey shaded area) time and

amplitude-normalised sEMG data from 9 and 8 trot strides are presented for left and right muscles, respectively. Coefficient of variation (CV) is indicated for each muscle.
(TIF)

**S2 Fig. Within-session (session 2) inter-subject sEMG profiles from the left and right-side muscles across the group of horses (n = 8).** Mean (solid line) and standard deviation (grey shaded area) time and amplitude-normalised sEMG data from 74 and 67 trot strides are presented for left and right muscles, respectively. Coefficient of variation (CV) is indicated for each muscle.
(TIF)

**S3 Fig. Within-session (session 1) intra-subject sEMG profiles from the left and right-side muscles of horse 1.** Mean (solid line) and standard deviation (grey shaded area) time and amplitude-normalised sEMG data from 10 trot strides are presented for each muscle. Coefficient of variation (CV) is indicated for each muscle.
(TIF)

**S4 Fig. Within-session (session 1) intra-subject sEMG profiles from the left and right-side muscles of horse 2.** Mean (solid line) and standard deviation (grey shaded area) time and amplitude-normalised sEMG data from 10 trot strides are presented for each muscle. Coefficient of variation (CV) is indicated for each muscle.
(TIF)

**S5 Fig. Within-session (session 1) intra-subject sEMG profiles from the left and right-side muscles of horse 3.** Mean (solid line) and standard deviation (grey shaded area) time and amplitude-normalised sEMG data from 10 trot strides are presented for each muscle. Coefficient of variation (CV) is indicated for each muscle.
(TIF)

**S6 Fig. Within-session (session 1) intra-subject sEMG profiles from the left-side muscles of horse 4.** Mean (solid line) and standard deviation (grey shaded area) time and amplitude-normalised sEMG data from 10 trot strides are presented for each muscle. Coefficient of variation (CV) is indicated for each muscle.
(TIF)

**S7 Fig. Within-session (session 1) intra-subject sEMG profiles from the left and right-side muscles of horse 5.** Mean (solid line) and standard deviation (grey shaded area) time and amplitude-normalised sEMG data from 9 and 10 trot strides are presented for left and right muscles, respectively. Coefficient of variation (CV) is indicated for each muscle.
(TIF)

**S8 Fig. Within-session (session 1) intra-subject sEMG profiles from the left and right-side muscles of horse 6.** Mean (solid line) and standard deviation (grey shaded area) time and amplitude-normalised sEMG data from 9 and 10 trot strides are presented for left and right muscles, respectively. Coefficient of variation (CV) is indicated for each muscle.
(TIF)

**S9 Fig. Within-session (session 1) intra-subject sEMG profiles from the left and right-side muscles of horse 7.** Mean (solid line) and standard deviation (grey shaded area) time and amplitude-normalised sEMG data from 10 trot strides are presented each muscle. Coefficient of variation (CV) is indicated for each muscle.
(TIF)

**S10 Fig. Within-session (session 1) intra-subject sEMG profiles from the left and right-side muscles of horse 8.** Mean (solid line) and standard deviation (grey shaded area) time and amplitude-normalised sEMG data from 10 trot strides are presented each muscle. Coefficient of variation (CV) is indicated for each muscle.
(TIF)

**S11 Fig. Within-session (session 1) inter-subject sEMG profiles from the left-side muscles across the group of horses (n = 8).** Mean (solid line) and standard deviation (grey shaded area) time and amplitude-normalised sEMG data from 78 trot strides are presented. Coefficient of variation (CV) is indicated for each muscle.
(TIF)

**S12 Fig. Within-session (session 2) intra-subject sEMG profiles from the left and right-side muscles of horse 1.** Mean (solid line) and standard deviation (grey shaded area) time and amplitude-normalised sEMG data from 10 trot strides are presented each muscle. Coefficient of variation (CV) is indicated for each muscle.
(TIF)

**S13 Fig. Within-session (session 2) intra-subject sEMG profiles from the left and right-side muscles of horse 2.** Mean (solid line) and standard deviation (grey shaded area) time and amplitude-normalised sEMG data from 8 and 6 trot strides are presented for left and right muscles, respectively. Coefficient of variation (CV) is indicated for each muscle.
(TIF)

**S14 Fig. Within-session (session 2) intra-subject sEMG profiles from the left and right-side muscles of horse 3.** Mean (solid line) and standard deviation (grey shaded area) time and amplitude-normalised sEMG data from 7 and 6 trot strides are presented for left and right muscles, respectively. Coefficient of variation (CV) is indicated for each muscle.
(TIF)

**S15 Fig. Within-session (session 2) intra-subject sEMG profiles from the left and right-side muscles of horse 5.** Mean (solid line) and standard deviation (grey shaded area) time and amplitude-normalised sEMG data from 10 and 8 trot strides are presented for left and right muscles, respectively. Coefficient of variation (CV) is indicated for each muscle.
(TIF)

**S16 Fig. Within-session (session 2) intra-subject sEMG profiles from the left and right-side muscles of horse 6.** Mean (solid line) and standard deviation (grey shaded area) time and amplitude-normalised sEMG data from 10 and 9 trot strides are presented for left and right muscles, respectively. Coefficient of variation (CV) is indicated for each muscle.
(TIF)

**S17 Fig. Within-session (session 2) intra-subject sEMG profiles from the left and right-side muscles of horse 7.** Mean (solid line) and standard deviation (grey shaded area) time and amplitude-normalised sEMG data from 10 trot strides are presented for each muscle. Coefficient of variation (CV) is indicated for each muscle.
(TIF)

**S18 Fig. Within-session (session 2) intra-subject sEMG profiles from the left and right-side muscles of horse 8.** Mean (solid line) and standard deviation (grey shaded area) time and amplitude-normalised sEMG data from 10 trot strides are presented for each muscle. Coefficient of variation (CV) is indicated for each muscle.
(TIF)

**S19 Fig. Between-session (session 1 and 2) intra-subject sEMG profiles from the left and right-side muscles of horse 1.** Mean (solid line) and standard deviation (grey shaded area) time and amplitude-normalised sEMG data from 20 trot strides are presented for each muscle. Coefficient of variation (CV) and coefficient of multiple correlation (CMC) is indicated for each muscle.
(TIF)

**S20 Fig. Between-session (session 1 and 2) intra-subject sEMG profiles from the left and right-side muscles of horse 2.** Mean (solid line) and standard deviation (grey shaded area) time and amplitude-normalised sEMG data from 18 and 16 trot strides are presented for left and right muscles, respectively. Coefficient of variation (CV) and coefficient of multiple correlation (CMC) is indicated for each muscle.
(TIF)

**S21 Fig. Between-session (session 1 and 2) intra-subject sEMG profiles from the left and right-side muscles of horse 3.** Mean (solid line) and standard deviation (grey shaded area) time and amplitude-normalised sEMG data from 17 and 16 trot strides are presented for left and right muscles, respectively. Coefficient of variation (CV) and coefficient of multiple correlation (CMC) is indicated for each muscle.
(TIF)

**S22 Fig. Between-session (session 1 and 2) intra-subject sEMG profiles from the left-side muscles of horse 4.** Mean (solid line) and standard deviation (grey shaded area) time and amplitude-normalised sEMG data from 19 trot strides are presented for each muscle. Coefficient of variation (CV) and coefficient of multiple correlation (CMC) is indicated for each muscle.
(TIF)

**S23 Fig. Between-session (session 1 and 2) intra-subject sEMG profiles from the left and right-side muscles of horse 5.** Mean (solid line) and standard deviation (grey shaded area) time and amplitude-normalised sEMG data from 19 and 18 trot strides are presented for left and right muscles, respectively. Coefficient of variation (CV) and coefficient of multiple correlation (CMC) is indicated for each muscle.
(TIF)

**S24 Fig. Between-session (session 1 and 2) intra-subject sEMG profiles from the left and right-side muscles of horse 6.** Mean (solid line) and standard deviation (grey shaded area) time and amplitude-normalised sEMG data from 19 trot strides are presented for each muscle. Coefficient of variation (CV) and coefficient of multiple correlation (CMC) is indicated for each muscle.
(TIF)

**S25 Fig. Between-session (session 1 and 2) intra-subject sEMG profiles from the left and right-side muscles of horse 7.** Mean (solid line) and standard deviation (grey shaded area) time and amplitude-normalised sEMG data from 20 trot strides are presented for each muscle. Coefficient of variation (CV) and coefficient of multiple correlation (CMC) is indicated for each muscle.
(TIF)

**S26 Fig. Between-session (session 1 and 2) intra-subject sEMG profiles from the left and right-side muscles of horse 8.** Mean (solid line) and standard deviation (grey shaded area)

time and amplitude-normalised sEMG data from 20 trot strides are presented for each muscle. Coefficient of variation (CV) is indicated for each muscle.
(TIF)

**S27 Fig. Between-session (session 1 and 2) inter-subject sEMG profiles from the left-side muscles across the group of horses (n = 8).** Mean (solid line) and standard deviation (grey shaded area) time and amplitude-normalised sEMG data from 152 trot strides are presented for each muscle. Coefficient of variation (CV) and coefficient of multiple correlation (CMC) is indicated for each muscle.
(TIF)

## Acknowledgments

The authors want to acknowledge Suzan Büchli BSc, who assisted with the preparation and execution of the study.

## Author Contributions

**Conceptualization:** L. St. George, T. J. P. Spoormakers, S. H. Roy, S. J. Hobbs, H. M. Clayton, F. M. Serra Bragança.

**Data curation:** L. St. George, T. J. P. Spoormakers, F. M. Serra Bragança.

**Formal analysis:** L. St. George, F. M. Serra Bragança.

**Funding acquisition:** L. St. George.

**Investigation:** L. St. George, T. J. P. Spoormakers, F. M. Serra Bragança.

**Methodology:** L. St. George, T. J. P. Spoormakers, S. H. Roy, S. J. Hobbs, H. M. Clayton, J. Richards, F. M. Serra Bragança.

**Project administration:** L. St. George, T. J. P. Spoormakers, F. M. Serra Bragança.

**Resources:** L. St. George, T. J. P. Spoormakers, F. M. Serra Bragança.

**Software:** L. St. George, F. M. Serra Bragança.

**Supervision:** T. J. P. Spoormakers, S. H. Roy, S. J. Hobbs, H. M. Clayton, J. Richards, F. M. Serra Bragança.

**Validation:** L. St. George, T. J. P. Spoormakers, S. H. Roy, S. J. Hobbs, H. M. Clayton, J. Richards, F. M. Serra Bragança.

**Visualization:** L. St. George, T. J. P. Spoormakers.

**Writing – original draft:** L. St. George, T. J. P. Spoormakers.

**Writing – review & editing:** L. St. George, T. J. P. Spoormakers, S. H. Roy, S. J. Hobbs, H. M. Clayton, J. Richards, F. M. Serra Bragança.

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
