## [Decision Letter · Decision Letter 0]

26 May 2023

PONE-D-23-09566Reliability of surface electromyographic (sEMG) measures of equine axial and appendicular muscles during overground trot.PLOS ONE

Dear Dr. St. George,

Thank you for submitting your manuscript to PLOS ONE. After careful consideration, we feel that it has merit but does not fully meet PLOS ONE’s publication criteria as it currently stands. Therefore, we invite you to submit a revised version of the manuscript that addresses the points raised during the review process.

We look forward to receiving your revised manuscript.

Kind regards,

Tomoyoshi Komiyama, Ph.D

Academic Editor

PLOS ONE

“I have read the journal's policy and the authors of this manuscript have the following competing interests: author SR is employed by Delsys Inc., the manufacturers of the sEMG sensors used in this study. The remaining authors declare that no competing interests exist.”

Additional Editor Comments:

Dear Authors,

Your study aimed to clarify whether sEMG profiles can reliably describe fundamental muscle activity patterns for selected equine muscles within a test session for individual horses. You found that these profiles are more variable across horses and between sessions, suggesting that it is reasonable to use sEMG to objectively monitor the intra-individual activity of these muscles across multiple gait evaluation sessions at in-hand trot.

However, I think you should strengthen the reliability of these results by adding as much information as possible.

We thus have some questions and suggestions for the manuscript that you might consider.

I believe these comments will be very helpful in the revision of your study.

Tomoyoshi Komiyama

Reviewers' comments:

Reviewer's Responses to Questions

**Comments to the Author**

1. Is the manuscript technically sound, and do the data support the conclusions?

Reviewer #1: Yes

Reviewer #2: Yes

Reviewer #3: Yes

2. Has the statistical analysis been performed appropriately and rigorously? 

Reviewer #1: Yes

Reviewer #2: Yes

Reviewer #3: Yes

3. Have the authors made all data underlying the findings in their manuscript fully available?

Reviewer #1: Yes

Reviewer #2: Yes

Reviewer #3: Yes

4. Is the manuscript presented in an intelligible fashion and written in standard English?

Reviewer #1: Yes

Reviewer #2: Yes

Reviewer #3: Yes

5. Review Comments to the Author

Reviewer #1: This is a well-designed study. The introduction and discussion are very verbose and can be significantly shortened so that the reader does not get bogged down and miss the important points.

Introduction:

The introduction is too long. It should be made more concise so that it covers the main points (briefly what is known in horses and humans, what the deficits are, justification of this study/gap in knowledge, objective/hypothesis). Much of the information that is more specific can be used in the discussion.

Line 44: This sentence should have multiple references not just a single reference from a review article.

Materials and Methods

Please include a paragraph on the study design. There seems to be 2 data collection session. When did these occur (same day, different day)? Please include this.

Lines 128-132: Please delete these sentences. This can be referenced in the appropriate sections of the materials and methods.

Lines 135-136: What university herd (Utrecht, Central Lancashire?). Please add this information.

Line 136: How were horses managed during the study period? Stall, regular turn-out? How long was the study period? Was the study performed at one or both of the universities?

Line 140: Add a sentence that gives a brief description of methods so that the reader does not have to go read the references. For example, placement of the markers and sEMG sensors is important. Since this is a repeatability study, how did you ensure accurate replacement of markers/sensors?

Line 181: Consider changing the section heading; data processing is a little more accurate (data analysis may be confused with statistical analysis).

Line 184: Please include a little more information on how hindlimb impact was detected (what marker(s)).

Lines 190-192: What joints were examined and was the back studied as a singular segment or multiples? Please include more information.

Line 225: What software was used to calculate these variables?

Line 238: What summary statistics were calculated and presented?

Lines 240-257: This paragraph is not appropriate for the statistical analysis section. This can be used in the discussion. The statistics section should present what statistics were performed not a discussion of why or what they mean.

Lines 266-273, 300-308: Instead of making a list of where all the data is contained, present pertinent data and then reference the tables/figures.

Lines 334-335: The results section presents the results – does not interpret them. This sentence can be used in the discussion.

Discussion

Lines 365-368: Please don’t include sentences about what you are about to discuss. This is unnecessary. Please delete these sentences.

Lines 371-374: Please avoid writing sentences using “we”. You could state that these techniques have been previously used in humans and equine.

Lines 375-376: Please include a sentence or 2 about how your CVs compare to these reported ones.

Lines 421-422: Do you think that investigation of muscle activation/timing may be a good variable to examine? This was identified as a variable that could change in response to lameness/underwater treadmill exercise (King et al AVJR 2017).

Line 436: Please change “is influenced” to “may be influenced”.

Lines 452-458: Please delete these sentences. It is not necessary to describe this. Describe what others have identified and how this is related to what you found.

Lines 458-465: This section can be made more concise by not repeating the results. It seems that you found differences in muscle groups between left and right sides. Has this been seen by other groups? Could this be from asymmetry, sidedness of the horse, difference in sensor placement?

Line 466: This seems to be a new thought process (not left vs right). A new paragraph could be started here.

Lines 497-501: Please delete these sentences. Just state the limitations. The subheading is “limitations”.

Lines 501-503: This sentence can be shortened. You have a small sample size that is not representative of all horses everywhere.

Lines 560-578: Restate the conclusions. This can be done in a couple sentences. Don’t restate your hypotheses or your discussion. Do not include information on lameness as this is beyond the scope of this study.

Lines 573-575: Please delete this sentence

Reviewer #2: This is a well written paper with appropriate statistical analysis and minimal or no grammatical or spelling errors. The references no 13 and 14 are referred to in the materials and methods. They are not fully referenced as they are still online as at the time of your submission, but as these are required to be read so that the methods are understood, the full references for both these papers need to be included.

Reviewer #3: Summary

This manuscript aims to determine whether a typical profile of sEMG activity can be reliably described within- and between-horses for selected appendicular and axial superficial muscles during in-hand trot. As well as to determine the between-session reliability of these profiles. Increased knowledge of normative sEMG profiles could serve as a base for future aid in clinical decision-making. However, there is a general lack of knowledge about the reliability of equine sEMG profiles, so this manuscript is a useful addition to the literature. The study focus is on the within- and between horses reliability as well as the between- sessions reliability.

General comments

This is an interesting, overall very well written manuscript and useful contribution to the available literature regarding the potential use of sEMG in clinical decision-making. The design is well described and the results clearly laid out.

The paper could benefit from additional information regarding the clinical implications of the results, such as an elaboration of the cut-of values for CMS when choosing to use the method or not. If so, it would give potential users of the technique important information.

Abstract

This describes the study clearly.

Introduction

The rationale for the study is well described and put into the clinical context.

Methods

The methods are well described. Recommendations- to add the maximum period between sessions and to clarify if it was the same handler for all horses and all registrations.

Results

These are comprehensively reported. Recommendations- to clarify if the horses had any asymmetries before the induction of lameness.

Discussion

This is a reasonable discussion of the findings. However, it would be interesting to have some additional discussion regarding the selection of “Reference voluntary contraction” for other gaits than trot.

Conclusions

The conclusions are relevant and supported by the results. However, on line 570 it states that the study results provide “evidence for sEMG as a reliable measure of muscle activity”. Maybe one could add that the validity needs to be investigated in future studies.

References

These are considered comprehensive.

Editorial comments:

P2, l19. Is it possible to define the type of study?

P3, l52.Is it possible to define what type of mechanisms?

P5, l89. “groupaverage”- or group- average?

P5, l96. 2932, should be 29, 32?

P5, l97. “betweensession”- this is sometimes spelled as one word (with and without an s) and sometimes as between-session(s) in the manuscript (the same applies to some other words in the manuscript such as intra-subject, within-session(s) and for the abstract as well)

P6, l118. Inhand- or in-hand?

P10, l193. Commonmode- or common-mode?

P10, l 220. Betweensessions- two words?

P12, l267. Intrasubject- one or two words?

P12, l273. Intrasubject- one or two words?

P14, l 306. Betweensessions- two words?

P19, l426. Betweenhorses- two words?

P21, l467. Affects

P23, l5121 betweentest- two words?

P25, l565. Betweensession- two words?

6. PLOS authors have the option to publish the peer review history of their article (what does this mean?). If published, this will include your full peer review and any attached files.

Reviewer #1: No

Reviewer #2: No

Reviewer #3: No

---

## [Author Response · Author response to Decision Letter 0]

7 Jun 2023

We would like to thank the reviewers and editor for taking the time to review our manuscript and for providing such positive and constructive feedback. We have endeavoured to make all requested changes to the manuscript and feel that this has improved the quality of the work. All changes are highlighted as tracked changes within the resubmitted manuscript and we have responded to each of the reviewer/editor comments below. We would like to thank the reviewers and editor in advance for reviewing the resubmitted manuscript for publication within PLOS One. 

Reviewer #1: This is a well-designed study. The introduction and discussion are very verbose and can be significantly shortened so that the reader does not get bogged down and miss the important points. Thank you for reviewing our manuscript and for your positive and constructive feedback.

Introduction:

The introduction is too long. It should be made more concise so that it covers the main points (briefly what is known in horses and humans, what the deficits are, justification of this study/gap in knowledge, objective/hypothesis). Much of the information that is more specific can be used in the discussion. Thank you, we have endeavoured to remove some information from this section to make it more concise. We do feel that much of the information provided is relevant for providing the reader with a sufficient background and justification for the study, so we were keen to maintain some of the detail from the original submission. We hope you will agree that our removal of some information is sufficient to mitigate your concerns. 

Line 44: This sentence should have multiple references not just a single reference from a review article. We have now included additional references for this sentence. 

Materials and Methods

Please include a paragraph on the study design. There seems to be 2 data collection session. When did these occur (same day, different day)? Please include this. Thank you for this suggestion. We have now included this information in lines 132 - 138. 

Lines 128-132: Please delete these sentences. This can be referenced in the appropriate sections of the materials and methods. This has now been deleted from the manuscript. 

Lines 135-136: What university herd (Utrecht, Central Lancashire?). Please add this information. Apologies for this omission. We have now specified that horses were from Utrecht University’s herd. 

Line 136: How were horses managed during the study period? Stall, regular turn-out? How long was the study period? Was the study performed at one or both of the universities? Thank you for this suggestion. We have provided further detail on the management of the horses during the study period and the length of the study period in lines 143 - 147. We have clarified that the study was performed at Utrecht University in line 132 - 133. 

Line 140: Add a sentence that gives a brief description of methods so that the reader does not have to go read the references. For example, placement of the markers and sEMG sensors is important. Since this is a repeatability study, how did you ensure accurate replacement of markers/sensors? Thank you for this suggestion. We have now included details on how we mitigated sensor re-application errors in lines 166 - 169. With respect, we would prefer to refer the reader to our previous studies which describe, in detail, sEMG sensor and retro-reflective marker locations. Repeating these locations in this study would substantially increase the length of the methods section and reviewers 2 and 3 have not requested this information, so we hope you will agree that it is sufficient to refer to our previous studies. 

Line 181: Consider changing the section heading; data processing is a little more accurate (data analysis may be confused with statistical analysis). Thank you for this suggestion. The section heading has now been changed to “Data Processing”.

Line 184: Please include a little more information on how hindlimb impact was detected (what marker(s)). In addition to referencing the method by Roepstorff et al., we have stated that the maximum vertical displacement of the marker placed between the tubera sacrale was used for stride segmentation (line 200 - 201). 

Lines 190-192: What joints were examined and was the back studied as a singular segment or multiples? Please include more information. We have now provided further information on the joint/segment angles in lines 205 - 212. 

Line 225: What software was used to calculate these variables? Apologies for this omission. sEMG profiles were calculated in Visual3D and this has now been added to the manuscript. 

Line 238: What summary statistics were calculated and presented? Thank you for noticing this omission. We have now specified that the mean ± standard deviation were calculated as summary statistics. 

Lines 240-257: This paragraph is not appropriate for the statistical analysis section. This can be used in the discussion. The statistics section should present what statistics were performed not a discussion of why or what they mean. We agree and have now moved parts of this paragraph to the discussion section and have left only the relevant sentences that relate to the methods used. 

Lines 266-273, 300-308: Instead of making a list of where all the data is contained, present pertinent data and then reference the tables/figures. Thank you for pointing this out. We have removed our description of the tables within these paragraphs and have instead referenced them where the associated data are described. However, we have not removed the sentences describing the figures and the data that these contain, as they present data from individual/representative horses, as well as from the group of horses (but only the right-side muscles in these instances). As such, these figures do not contain data from the entire dataset and therefore cannot be referenced in relation to the results that are presented in the text. We have shortened these sentences, so that they take up less space within the paragraphs but hope you will agree that it is helpful for the reader to have these descriptions of the Figures within the text. 

Lines 334-335: The results section presents the results – does not interpret them. This sentence can be used in the discussion. Thank you for this suggestion. We have altered the sentence to remove the comparison with findings for within-session reliability. We hope you will agree that the revised sentence only presents the results. 

Discussion

Lines 365-368: Please don’t include sentences about what you are about to discuss. This is unnecessary. Please delete these sentences. With respect, we would prefer to keep this sentence, as some readers may expect a discussion of sEMG activity profiles in relation to muscle function, particularly within the context of the joint/segment motion during trot. We feel it is important to state that our discussion will instead focus on the reliability of the sEMG profiles, and that the reader can refer to the “sister” manuscripts where the above information are described in detail. We hope you will agree with our reasons for including this sentence at the outset of the discussion section. 

Lines 371-374: Please avoid writing sentences using “we”. You could state that these techniques have been previously used in humans and equine. Thank you, this has now been changed. 

Lines 375-376: Please include a sentence or 2 about how your CVs compare to these reported ones. With respect, we would prefer not to make this comparison, as these CV thresholds have been used for discrete data only and are therefore not directly applicable to the continuous data that were used in our study. We explain this in the following sentence and make the suggested comparison with CV values that have been reported in other studies that employed comparable continuous data. As such, we hope you will agree that making such a comparison with the values in the highlighted lines would not be appropriate and may confuse the reader. 

Lines 421-422: Do you think that investigation of muscle activation/timing may be a good variable to examine? This was identified as a variable that could change in response to lameness/underwater treadmill exercise (King et al AVJR 2017). Yes, we agree that this would be a relevant variable to study using the CV in future studies. We also noted that discrete onset/offset variables were significantly altered during induced fore- and hindlimb lameness in St. George et al. (2022) and Spoormakers et al. (2023). However, we felt that an examination of these variables went beyond the scope of this preliminary evaluation. 

Line 436: Please change “is influenced” to “may be influenced”. This has now been changed. 

Lines 452-458: Please delete these sentences. It is not necessary to describe this. Describe what others have identified and how this is related to what you found. We have now removed this introductory sentence.

Lines 458-465: This section can be made more concise by not repeating the results. It seems that you found differences in muscle groups between left and right sides. Has this been seen by other groups? Could this be from asymmetry, sidedness of the horse, difference in sensor placement? Thank you, we have now removed the results from these sentences and have included an additional sentence to touch on the bilateral differences in CMC that were observed for biceps femoris. We are not aware of other studies that have observed this in horses during trot. 

Line 466: This seems to be a new thought process (not left vs right). A new paragraph could be started here. Thank you, we agree and have now created a new paragraph. 

Lines 497-501: Please delete these sentences. Just state the limitations. The subheading is “limitations”. These sentences have now been deleted. 

Lines 501-503: This sentence can be shortened. You have a small sample size that is not representative of all horses everywhere. This sentence has been revised. 

Lines 560-578: Restate the conclusions. This can be done in a couple sentences. Don’t restate your hypotheses or your discussion. Do not include information on lameness as this is beyond the scope of this study. We have now made this section more concise and have removed the below sentence regarding lameness but would prefer to leave our mention of lameness in the concluding sentence, as this is included in the introduction as a reason for conducting this study and it highlights the potential impact of our findings. 

Lines 573-575: Please delete this sentence. This sentence has been removed. 

Reviewer #2: This is a well written paper with appropriate statistical analysis and minimal or no grammatical or spelling errors. The references no 13 and 14 are referred to in the materials and methods. They are not fully referenced as they are still online as at the time of your submission, but as these are required to be read so that the methods are understood, the full references for both these papers need to be included. Thank you for reviewing our manuscript and for your positive feedback. We have now updated reference 14 to include the volume and issue number, which have since been released since the initial submission of this manuscript. Unfortunately, Spoormakers et al. [13] is still published online as an “Early View” article (Online Version of Record before inclusion in an issue) in EVJ, so we cannot update the record further in this resubmission. We have, however, included the doi for this reference to ensure that it is accessible for the reader. 

Reviewer #3: Summary: This manuscript aims to determine whether a typical profile of sEMG activity can be reliably described within- and between-horses for selected appendicular and axial superficial muscles during in-hand trot. As well as to determine the between-session reliability of these profiles. Increased knowledge of normative sEMG profiles could serve as a base for future aid in clinical decision-making. However, there is a general lack of knowledge about the reliability of equine sEMG profiles, so this manuscript is a useful addition to the literature. The study focus is on the within- and between horses reliability as well as the between- sessions reliability. Thank you for taking the time to review our manuscript and for providing such constructive feedback. We have endeavoured to make all requested within the manuscript and have addressed each of your comments below.

General comments

This is an interesting, overall very well written manuscript and useful contribution to the available literature regarding the potential use of sEMG in clinical decision-making. The design is well described and the results clearly laid out. 

The paper could benefit from additional information regarding the clinical implications of the results, such as an elaboration of the cut-of values for CMS when choosing to use the method or not. If so, it would give potential users of the technique important information. Thank you for your constructive feedback and suggestions for improvement. Unfortunately, the preliminary nature of this study and the relatively small sample size means that we are not able to define or comment on CV or CMC cut-off values. We do agree that this is important information for using sEMG to evaluate muscle activity in horses and we have commented on the need for future studies to build on the results presented here, so that these cut-offs can be determined (lines 468 – 471, 590 – 592, 609 - 612). We hope you will agree that highlighting this requirement is sufficient for this preliminary study. 

Abstract: This describes the study clearly. Thank you! 

Introduction: The rationale for the study is well described and put into the clinical context. Thank you! 

Methods: The methods are well described. Recommendations- to add the maximum period between sessions and to clarify if it was the same handler for all horses and all registrations. Thank you for these suggestions. We have now included the maximum period between sessions in lines 137 - 138 of the revised manuscript. In line 187, it is stated that one handler led the horses throughout the data collection sessions, but we have tried to clarify this. 

Results: These are comprehensively reported. Recommendations- to clarify if the horses had any asymmetries before the induction of lameness. Thank you for the positive feedback on this section. We confirm in lines 143 – 144 and 190 – 194 that horses were deemed as clinically non-lame by two experienced equine veterinarians prior to, and during, the study. With respect, we would prefer not to report on asymmetries in the results section, as we are only reporting data from the baseline, or “non-lame”, condition and the potential impact of movement asymmetry on the results are described in detail in the discussion section (lines 539 - 545). We hope you will agree that this is the most appropriate way of addressing this. 

Discussion: This is a reasonable discussion of the findings. However, it would be interesting to have some additional discussion regarding the selection of “Reference voluntary contraction” for other gaits than trot. Thank you for the positive feedback on this section. We would like to draw the reviewer’s attention to lines 569 - 581 within the limitations section, where we discuss the normalisation method used. The RVC employed for normalisation (the peak amplitude observed across trot strides within each horse, muscle, and test session) could be applied to sEMG signals, irrespective of the gait studied. With respect, we would prefer not to include this in our existing discussion of normalisation, as we feel it would disrupt the flow and may be confusing for some readers. We hope you will agree with this. 

Conclusions: The conclusions are relevant and supported by the results. However, on line 570 it states that the study results provide “evidence for sEMG as a reliable measure of muscle activity”. Maybe one could add that the validity needs to be investigated in future studies. Thank you for this suggestion. We feel we’ve addressed the need for future validation studies in lines 609 - 612, where we state that “future research should build on this preliminary study to explore, and eventually validate, normative sEMG profiles for equine gait …”. We hope you will agree that this sentence addresses your suggestion for improving this section. 

References: These are considered comprehensive. Thank you!

Editorial comments:

P2, l19. Is it possible to define the type of study? We have now included “observational” study within this sentence. 

P3, l52.Is it possible to define what type of mechanisms? Apologies, we have now added “neuromuscular mechanisms” for clarity. 

P5, l89. “groupaverage”- or group- average? We can confirm that this is correctly written as “group-averaged”.

P5, l96. 2932, should be 29, 32? We can confirm that this is referenced correctly as “29 – 32”.

P5, l97. “betweensession”- this is sometimes spelled as one word (with and without an s) and sometimes as between-session(s) in the manuscript (the same applies to some other words in the manuscript such as intra-subject, within-session(s) and for the abstract as well). Thank you for pointing this out. We hyphenate “between-session” when we refer to the form of reliability that we studied, but we do not hyphenate “between sessions” when we are describing the two separate data collection sessions that were conducted. However, we have endeavoured to make this more consistent throughout the manuscript as tracked changes. 

P6, l118. Inhand- or in-hand? We can confirm that this is correctly written as “in-hand” 

P10, l193. Commonmode- or common-mode? We can confirm that this is correctly written as “common-mode”

P10, l 220. Betweensessions- two words? We can confirm that this is correctly written as “between-session”

P12, l267. Intrasubject- one or two words? We can confirm that this is correctly written as “intra-subject”

P12, l273. Intrasubject- one or two words? We can confirm that this is correctly written as “intra-subject”

P14, l 306. Betweensessions- two words? We can confirm that this is correctly written as “between-session”

P19, l426. Betweenhorses- two words? Thank you for noticing this. We have now written this as two words (between horses). 

P21, l467. Affects Thank you for noticing this, which we have now corrected.

P23, l5121 betweentest- two words? ? Thank you for noticing this. We have now written this as two words (between test). 

P25, l565. Betweensession- two words? Thank you for noticing this. We have now written this as two words (between horses).

---

## [Decision Letter · Decision Letter 1]

14 Jun 2023

PONE-D-23-09566R1Reliability of surface electromyographic (sEMG) measures of equine axial and appendicular muscles during overground trot.PLOS ONE

Dear Dr. St. George,

Thank you for submitting your manuscript to PLOS ONE. After careful consideration, we feel that it has merit but does not fully meet PLOS ONE’s publication criteria as it currently stands. Therefore, we invite you to submit a revised version of the manuscript that addresses the points raised during the review process.

We look forward to receiving your revised manuscript.

Kind regards,

Tomoyoshi Komiyama, Ph.D

Academic Editor

PLOS ONE

Journal Requirements:

Additional Editor Comments:

Dear authors,

Thank you for your submitting your revised manuscript.

I think it is easier to understand than the previous version.

However, Reviewer 1 had additional comments.

Please answer these questions as listed below.

Tomoyoshi Komiyama

Reviewers' comments:

Reviewer's Responses to Questions

**Comments to the Author**

1. If the authors have adequately addressed your comments raised in a previous round of review and you feel that this manuscript is now acceptable for publication, you may indicate that here to bypass the “Comments to the Author” section, enter your conflict of interest statement in the “Confidential to Editor” section, and submit your "Accept" recommendation.

Reviewer #1: (No Response)

Reviewer #3: All comments have been addressed

2. Is the manuscript technically sound, and do the data support the conclusions?

Reviewer #1: Yes

Reviewer #3: Yes

3. Has the statistical analysis been performed appropriately and rigorously? 

Reviewer #1: Yes

Reviewer #3: Yes

4. Have the authors made all data underlying the findings in their manuscript fully available?

Reviewer #1: Yes

Reviewer #3: Yes

5. Is the manuscript presented in an intelligible fashion and written in standard English?

Reviewer #1: Yes

Reviewer #3: Yes

6. Review Comments to the Author

Reviewer #1: The authors have done a nice job editing the manuscript based on previous recommendations. I have a few additional comments:

Introduction:

While there is good information within the introduction, it is still too long. Because of this, the reader will miss the key details and be stuck in the weeds of the less relevant information.

In paragraph 1, lines 53-54 is not important background or justification for the study and can be deleted. Most of paragraph 2 is not relevant and can be condensed to several sentences. It seems that the main point of this paragraph is that expected variability of sEMG in normal horses needs to be determined. Lines 73-82 seems unnecessary. Specifics of human studies (such as the gaits that have been evaluated) are not necessary to make your point (lines 83-85); this sentence can be deleted since the relevant finding to your argument is in the next sentence.

Lines 88-92: the first part of the sentence is good (you lose valuable information when you group average data). Please remove the second part of the sentence starting at “so it has been suggested…”

Lines 95-102: none of this adds to your argument. Please delete.

Lines 103-104: It is worthwhile to state that treadmill locomotion and over-ground locomotion are not the same. As most horses will not be evaluated when on a treadmill, it seems relevant to examine this overground. This would be a gap in knowledge that you would fill with this project.

It is not necessary to have references for your hypothesis. Please remove “Based on previous sEMG literature from humans and equine subjects” (lines 109-110). Just state your hypothesis!

Materials and Methods

Line 143: What size clipper blade was used?

Line 145: Was the sensor placed over the middle/center of the muscle belly? Please include this information.

Results:

Lines 247-248: I assume that this number of strides is for both forelimbs and hindlimbs.

Lines 254-268: It is fine to reference the representative figures; however, present the pertinent results first and move the representative figures to the end of the paragraph. Start with line 260 (Across test sessions and muscles …)

Lines 287-297: Same as the comment above (present the results first followed by the representative images).

Discussion:

Lines 351-354: This sentence is not necessary. Please delete.

Lines 370-374: Instead of stating that there are no studies to report CVs in equine, you can just state that this is the first study to report these values in horses.

Lines 375-377: This sentence is a bit confusing. I don’t think you need the statement “across all muscles and test sessions.

Lines 377-379: Was this data from the same data collection session as the current study? If so, this should be deleted.

Lines 388-392: This is a very long and confusing sentence. This sentence could be separated into 2 which may help. There seem to be multiple trains of thought in this sentence (force impulse and electrode-skin interface). Please revise.

Lines 394-397: These 2 sentences state the same thing. Delete the first sentence (Small mean values…).

Lines 426-427: Do you mean that you did not see the same magnitude of disparity? Please clarify.

Line 457: Please replace “is” with “was”

Reviewer #3: Thank you for revising the manuscript and in for some comments, answering why you did not. I have no further comments.

7. PLOS authors have the option to publish the peer review history of their article (what does this mean?). If published, this will include your full peer review and any attached files.

Reviewer #1: No

Reviewer #3: No

---

## [Author Response · Author response to Decision Letter 1]

15 Jun 2023

We would like to thank the reviewers and editor for taking the time to review our manuscript in this second round and for providing such positive and constructive feedback. We have endeavoured to make all requested changes to the manuscript and feel that this has improved the quality of the work. All changes are highlighted as tracked changes within the resubmitted manuscript and we have responded to each of the reviewer/editor comments below in blue text. We would like to thank the reviewers and editor in advance for reviewing the resubmitted manuscript for publication within PLOS One. 

Reviewer #1: The authors have done a nice job editing the manuscript based on previous recommendations. I have a few additional comments: Thank you for reviewing our manuscript and for your positive and constructive feedback.

Introduction:

While there is good information within the introduction, it is still too long. Because of this, the reader will miss the key details and be stuck in the weeds of the less relevant information. Thank you for providing suggestions for how we can further reduce the length of this section. We have endeavoured to make all requested changes and hope you will agree that it is now a more appropriate length that is suitable for publication. 

In paragraph 1, lines 53-54 is not important background or justification for the study and can be deleted. This sentence has now been deleted. 

Most of paragraph 2 is not relevant and can be condensed to several sentences. It seems that the main point of this paragraph is that expected variability of sEMG in normal horses needs to be determined. Lines 73-82 seems unnecessary. Thank you, lines 73 – 82 have now been deleted and we hope you will agree that this paragraph is now sufficiently condensed. 

Specifics of human studies (such as the gaits that have been evaluated) are not necessary to make your point (lines 83-85); this sentence can be deleted since the relevant finding to your argument is in the next sentence. This sentence has now been deleted. 

Lines 88-92: the first part of the sentence is good (you lose valuable information when you group average data). Please remove the second part of the sentence starting at “so it has been suggested…” Thank you, we have removed this portion of the sentence.

Lines 95-102: none of this adds to your argument. Please delete. This has now been deleted.

Lines 103-104: It is worthwhile to state that treadmill locomotion and over-ground locomotion are not the same. As most horses will not be evaluated when on a treadmill, it seems relevant to examine this overground. This would be a gap in knowledge that you would fill with this project. Thank you, we have now included a sentence in lines 118 – 120 to cover this point. 

It is not necessary to have references for your hypothesis. Please remove “Based on previous sEMG literature from humans and equine subjects” (lines 109-110). Just state your hypothesis! Thank you, this has now been removed. 

Materials and Methods 

Line 143: What size clipper blade was used? Apologies for this omission, we have now included details on the clipper blade in line 163.

Line 145: Was the sensor placed over the middle/center of the muscle belly? Please include this information. Apologies for this omission, we have now clarified that the sensor was placed over the middle of the muscle belly in line 166.

Results:

Lines 247-248: I assume that this number of strides is for both forelimbs and hindlimbs. Yes, it is inclusive of all muscles studied (from the fore- and hindlimbs, and longissimus). 

Lines 254-268: It is fine to reference the representative figures; however, present the pertinent results first and move the representative figures to the end of the paragraph. Start with line 260 (Across test sessions and muscles …) Thank you for this suggestion, we have now moved these sentences to the end of the paragraph.

Lines 287-297: Same as the comment above (present the results first followed by the representative images). Thank you, the suggested changes have been implemented, as above. 

Discussion:

Lines 351-354: This sentence is not necessary. Please delete. This has now been deleted. 

Lines 370-374: Instead of stating that there are no studies to report CVs in equine, you can just state that this is the first study to report these values in horses. Thank you, this has now been amended. 

Lines 375-377: This sentence is a bit confusing. I don’t think you need the statement “across all muscles and test sessions. Thank you, we have now deleted this statement from the sentence. 

Lines 377-379: Was this data from the same data collection session as the current study? If so, this should be deleted. Yes, it was, so we have now deleted this sentence. 

Lines 388-392: This is a very long and confusing sentence. This sentence could be separated into 2 which may help. There seem to be multiple trains of thought in this sentence (force impulse and electrode-skin interface). Please revise. Thank you, we have now split this into two sentences for clarity. 

Lines 394-397: These 2 sentences state the same thing. Delete the first sentence (Small mean values…). Thank you, we have now deleted the first sentence. 

Lines 426-427: Do you mean that you did not see the same magnitude of disparity? Please clarify. Apologies, we intended to state that our overall finding of higher CVs for inter- vs. intra-subject sEMG profiles agrees with the human literature. We have amended this sentence for clarity. 

Line 457: Please replace “is” with “was” Thank you for noticing this error, which we have now corrected. 

Reviewer #3: Thank you for revising the manuscript and in for some comments, answering why you did not. I have no further comments. Thank you for reviewing our revised manuscript and for your positive and constructive feedback.

---

## [Decision Letter · Decision Letter 2]

2 Jul 2023

Reliability of surface electromyographic (sEMG) measures of equine axial and appendicular muscles during overground trot.

PONE-D-23-09566R2

Dear Dr. St. George,

We’re pleased to inform you that your manuscript has been judged scientifically suitable for publication and will be formally accepted for publication once it meets all outstanding technical requirements.

Kind regards,

Tomoyoshi Komiyama, Ph.D

Academic Editor

PLOS ONE

Additional Editor Comments (optional):

Dear authors,

Thank you for submitting your revised manuscript.

It was much easier to understand than the original manuscript.

I am satisfied with the responses and edits made per the reviewers’ comments, therefore I am happy to accept you study.

Also, I believe this manuscript will satiate the reader's interest.

Tomoyoshi Komiyama

Reviewers' comments:

Reviewer's Responses to Questions

**Comments to the Author**

1. If the authors have adequately addressed your comments raised in a previous round of review and you feel that this manuscript is now acceptable for publication, you may indicate that here to bypass the “Comments to the Author” section, enter your conflict of interest statement in the “Confidential to Editor” section, and submit your "Accept" recommendation.

Reviewer #1: All comments have been addressed

2. Is the manuscript technically sound, and do the data support the conclusions?

Reviewer #1: Yes

3. Has the statistical analysis been performed appropriately and rigorously? 

Reviewer #1: Yes

4. Have the authors made all data underlying the findings in their manuscript fully available?

Reviewer #1: Yes

5. Is the manuscript presented in an intelligible fashion and written in standard English?

Reviewer #1: Yes

6. Review Comments to the Author

Reviewer #1: Thank you for your responses to previous comments. This manuscript is acceptable for publication.

7. PLOS authors have the option to publish the peer review history of their article (what does this mean?). If published, this will include your full peer review and any attached files.

Reviewer #1: No

---

## [Editor Report · Acceptance letter]

6 Jul 2023

PONE-D-23-09566R2 

Reliability of surface electromyographic (sEMG) measures of equine axial and appendicular muscles during overground trot. 

Dear Dr. St. George:

I'm pleased to inform you that your manuscript has been deemed suitable for publication in PLOS ONE. Congratulations! Your manuscript is now with our production department. 

Kind regards, 

on behalf of

Dr. Tomoyoshi Komiyama 

Academic Editor

PLOS ONE